# L-Glyceraldehyde Inhibits Neuroblastoma Cell Growth via a Multi-Modal Mechanism on Metabolism and Signaling

**DOI:** 10.3390/cancers16091664

**Published:** 2024-04-25

**Authors:** Martin Forbes, Richard Kempa, Guido Mastrobuoni, Liam Rayman, Matthias Pietzke, Safak Bayram, Birte Arlt, Annika Spruessel, Hedwig E. Deubzer, Stefan Kempa

**Affiliations:** 1Integrative Proteomics and Metabolomics, Berlin Institute for Medical Systems Biology, Max-Delbrück Center for Molecular Medicine in the Helmholtz Association, Hannoversche Str. 28, 10115 Berlin, Germany; 2Department of Pediatric Hematology and Oncology, Charité—Universitätsmedizin Berlin, Augustenburger Platz 1, 13353 Berlin, Germany; 3Mass Spectrometry Facility, MaxPlanck Institute for Molecular Genetics, Ihnestrasse 63-73, 14195 Berlin, Germany; 4Berliner Institut für Gesundheitsforschung (BIH), Anna-Louisa-Karsch-Strase 2, 10178 Berlin, Germany; 5German Cancer Consortium (DKTK), Partner Site Berlin, Invalidenstr. 80, 10115 Berlin, Germany; 6German Cancer Research Center (DKFZ), Im Neuenheimer Feld 280, 69120 Heidelberg, Germany; 7Experimental and Clinical Research Center (ECRC), Charité and Max-Delbrück-Center for Molecular Medicine (MDC) in the Helmholtz Association, 13125 Berlin, Germany

**Keywords:** cancer metabolism, redox mechanisms, nucleotide biosynthesis, glycolytic inhibition

## Abstract

**Simple Summary:**

Approximately half of all infants and children newly diagnosed with neuroblastoma are at high risk for relapse despite aggressive multi-modal therapy. Polychemotherapy usually provokes a good initial response; however, minimal residual disease with the dissemination of a few resistant tumor cells frequently causes relapse. The number of long-term survivors of high-risk disease has remained unsatisfactorily poor with survival rates as low as 50% after first-line therapy and below 20% in the relapse situation. Applying extensive state-of-the-art methods and workflows for metabolomics and proteomics, this study characterized the modes of action of the metabolic inhibitor L-glyceraldehyde in a panel of neuroblastoma cell lines as well as fibroblasts used as a control. Our analyses revealed that L-glyceraldehyde arrests proliferation and induces neuroblastoma cell death by a multi-modal mechanism that involves oxidative stress and the inhibition of nucleotide biosynthesis. Our data provide the molecular basis for the rationale design of multi-modal therapies involving metabolic targets.

**Abstract:**

Glyceraldehyde (GA) is a three-carbon monosaccharide that can be present in cells as a by-product of fructose metabolism. Bruno Mendel and Otto Warburg showed that the application of GA to cancer cells inhibits glycolysis and their growth. However, the molecular mechanism by which this occurred was not clarified. We describe a novel multi-modal mechanism by which the L-isomer of GA (L-GA) inhibits neuroblastoma cell growth. L-GA induces significant changes in the metabolic profile, promotes oxidative stress and hinders nucleotide biosynthesis. GC-MS and ^13^C-labeling was employed to measure the flow of carbon through glycolytic intermediates under L-GA treatment. It was found that L-GA is a potent inhibitor of glycolysis due to its proposed targeting of NAD(H)-dependent reactions. This results in growth inhibition, apoptosis and a redox crisis in neuroblastoma cells. It was confirmed that the redox mechanisms were modulated via L-GA by proteomic analysis. Analysis of nucleotide pools in L-GA-treated cells depicted a previously unreported observation, in which nucleotide biosynthesis is significantly inhibited. The inhibitory action of L-GA was partially relieved with the co-application of the antioxidant N-acetyl-cysteine. We present novel evidence for a simple sugar that inhibits cancer cell proliferation via dysregulating its fragile homeostatic environment.

## 1. Introduction

In 1929, Bruno Mendel reported that glyceraldehyde (GA) inhibits anaerobic fermentation in Jensen sarcoma without affecting respiration in tumor or normal rat tissue [1]. Glyceraldehyde is the non-phosphorylated form of the glycolysis intermediate, glyceraldehyde-3-phosphate. Five years previous to Mendel’s discovery, Otto Warburg found that cancer cells exhibit an addiction to glucose and show high glycolytic activity. Specifically, cancer cells have a preference towards glycolysis and the secretion of Lactate over oxidative phosphorylation even in the presence of oxygen [2]. This observation between the metabolism of non-cancerous and cancerous cells gives reasoning to the efficacy of GA as a therapeutic via glycolytic inhibition. This prompted the seminal research led by Warburg, showing that GA was effective in curing cancer ascites in all 75 mice that were treated [3]. Numerous laboratories conducted experiments with GA and reported varied efficacy, or none at all [4,5,6]. A driving factor in the contentious nature of GA efficacy is that, the D- and L- isomers of GA had differing potency [7]. The synthesis of pure L-GA demonstrated its superiority over D-GA in glycolytic inhibition [8]. Due to achieving only relative success in vivo, GA went largely out of fashion. One of the last published experiments on the efficacy of GA in vivo showed that neuroblastoma cells were sensitive to GA and exhibited inhibited glycolysis and cell proliferation [9]. However, the molecular mechanism by which GA inhibits cancer cells was not established. Later research focused solely on GA as a stimulant of insulin secretion in pancreatic cells, thereby shelving its potential as a therapeutic [10,11].

The rapid but inefficient production of ATP via glycolysis comes at the cost of increased oxidative state [12]. Cancer cells in proliferative states require a metabolic adaptation to accommodate the increase in reductive and oxidative cellular processes (redox). Reactive oxygen species (ROS) accumulate due to the increase in metabolic processes. The cancer cell requires mechanisms to detoxify the cell of the ROS and maintain a homeostatic redox balance [13]. This system is finely tuned to avoid ROS-induced autophagy and apoptosis [14]. By targeting ROS-sensitive metabolic processes, including glycolysis and nucleotide biosynthesis, opportunities for novel—or a renaissance of—therapeutics arise.

Neuroblastoma entails an extra-cranial solid tumor occurring in the developmental sympathetic nervous system. Of all childhood cancers, neuroblastoma accounts for 15%, making it the most commonly diagnosed malignancy in the first years of life [15]. Approximately 90% of neuroblastoma tumors are diagnosed in children under the age of 10, with a median age of 18 months [16]. In one high-risk group, long-term survival is below 50%. In addition, the treatment regimens are intense and multi-modal including polychemotherapy, surgery, high-dose chemotherapy and analogous stem cell transplantation, radiotherapy, and immunotherapy [17,18].

A common feature of *MYCN* amplified neuroblastoma cells is their high levels of ROS production [19]. Given that MYCN increases ROS production, compounds that further increase ROS may provide valid targets for therapeutics. As a pretext to the research presented here, it was found that GA had the potential for generating ROS in pancreatic islet cells [20]. There is scope to assess the function of L-GA in relation to ROS production and its efficacy in tipping the fine balance towards cancer cell death. We propose that L-GA acts in a multi-modal fashion on the metabolism to inhibit cancer cell growth. Here, we examine the effect of L-GA on metabolic and proteomic processes in neuroblastoma and fibroblast cells. The data allude to the generation of ROS, in combination with inhibitory action on the central carbon metabolism and nucleotide biosynthesis pathways.

## 2. Materials and Methods

Cell lines were maintained in DMEM (Thermo Fisher Scientific Waltham, MA, USA) medium, without glucose, glutamine, phenol red, and sodium pyruvate. The media were supplemented with 10% fetal bovine serum (Thermo Fisher Scientific Waltham, MA, USA), 2 mM glutamine (Thermo Fisher Scientific Waltham, MA, USA), and 2.5 g/L glucose (Merck, Darmstadt, Germany) and cultivated at 37 °C, 5% CO_2_, 21% O_2_ and 85% relative humidity. To avoid contact inhibition, cells were passaged every 3–4 days. The neuroblastoma cell lines BE(2)-C (RRID: CVCL_0529) were obtained from ECACC (Salisbury, UK). The IMR-32 (RRID: CV CL_0346), GI-ME-N (RRID: CVCL_1232) and SH-SY5Y (RRID: CVCL_0019) cell lines were purchased from the DSMZ (Braunschweig, Germany). Cell lines were authenticated via the Multiplex human Cell line Authentication Test (Multiplexion, Immenstaad, Germany). The active primary human foreskin fibroblasts from an infant donor (VH7) were a gift from Petra Boukamp (German Cancer Research Center (DKFZ), Heidelberg, Germany). L-glyceraldehyde (73572, Sigma-Aldrich, St. Louis, MO, USA), D-glyceraldehyde (49800, Sigma-Aldrich, St. Louis, MO, USA) were prepared in 1M stock solutions in PBS and stored at 4% before use. N-acetyl-cysteine (A9165, Sigma-Aldrich, St. Louis, MO, USA) was prepared in a 500 mM solution in PBS fresh before use in cell culture.

### 2.1. Proliferation Assay

Numbers of viable cells were determined using trypan blue exclusion. A cell suspension was prepared with 50 μL 0.4% trypan blue solution (*w*/*v*) and 50 μL trypsinized cells. Following mixing, 10 μL of the cell suspension was loaded into a cell counting slide. Cell numbers were measured using the TC10 automated cell counter (Biorad, Hercules, CA, USA).

### 2.2. Proteomic Analysis

Cells were harvested by scratching cells in 2 mL ice-cold 1× PBS on ice and transferred to a 2 mL tube; cells were centrifuged at 14,000× *g* for 5 min, 4 °C. Cell pellets were lysed in 300 μL urea buffer (8 M Urea, 100 mM TrisHCl (pH 8.5). Lysates were sonicated for 30 s and centrifuged at 14,000× *g* for 5 min, 4 °C. Lysates were then incubated for 10 min at 4 °C under constant agitation. Lysates were centrifuged at 14,000× *g* for 5 min, 4 °C, supernatants were removed, and the protein concentration was determined via the Pierce BCA Protein Assay Kit (Thermo Fisher Scientific Waltham, MA, USA). The absorbance of each sample was read at a wavelength of 562 nm (Infinite 2000, Tecan, Kanagawa, Japan). Each sample was measured in technical duplicates.

Proteins were alkylated and denatured by the addition of 2 mM DTT for 30 min at 25 °C, followed by the addition of 11 mM iodoacetamide for 15 min at room temperature in the dark. One hundred μg of protein was aliquoted and digested using Lys-C (Wako, Hiroshima-shi, Japan) 1:40 (*w*/*w*) and immobilized trypsin beads (5–10 μL, 4 h under rotation, 30 °C). Digested proteins were diluted with 50 mM ammonium bicarbonate before tryptic digestion. Digestion was halted with 5 μL of triflouracetic acid (TFA). Then, 20 μg of the digested sample was desalted and purified on in-house prepared stage tips. Stages tips were primed with 50 μL 100% methanol and 50 μL 0.5% acetic acid (*v*/*v*). Stage tips were then centrifuged at 300× *g* for 7 min, and digests were loaded into the tips and washed once with 50 μL 0.5% acetic acid (*v*/*v*). Peptides were eluted by the addition of 10 μL 0.5% acetic acid (*v*/*v*) in 80% acetonitrile. Eluates were dried by centrifugation under vacuum then resuspended in 10 μL 0.5% acetic acid (*v*/*v*) and sonicated at room temperature for 5 min. For phosphoproteomic analysis, digested samples were desalted using SepPak C18 columns prior to phosphopeptide enrichment using the High-Select TiO2 Phosphopeptide Enrichment Kit (Thermo Fisher Scientific Waltham, MA, USA). Samples were resuspended in 150 μL of Binding/Equilibration Buffer and vortexed. Samples were then enriched for phosphopeptides using a TiO2 Spin Tip, washed and eluted in preparation for LC-MS.

### 2.3. Analysis of Peptides via Liquid Chromatography–Mass Spectrometry (LC-MS)

Peptide mixtures were analyzed by LC-MS following a shotgun proteomics method [21]. LC-MS was performed on an automated high-performance liquid-chromatograph (NanoLC 415, Eksigent, Dublin, CA, USA) coupled to tandem mass spectrometry (Q Exactive HF, Thermo Fisher Scientific Waltham, MA, USA). Samples were pipetted into 5 μL aliquots into a 96-well plate to be acquired in two technical replicates. Samples were loaded on the column with a flow rate of 450 nL/min. Elution was performed with a flow rate of 400 nL/min using a 240 min gradient ranging from 5% to 40% of buffer B (80% acetonitrile and 0.1% formic acid) in buffer A (5% acetonitrile in 0.1% formic acid). Chromatographic separation was performed on a 200 cm long MonoCap C18 High Resolution 2000 column (GL Scientific, Shinjuku-ku, Janpan). The nanospray source of the Q Exactive HF was maintained at 2.4 kV and the downstream ion transfer tube at 260 °C. Data were acquired in a data-dependent mode with one survey MS scan (resolution: R = 120,000 at *m*/*z* 200) followed by a maximum of ten MS/MS scans (resolution: R = 30,000 at *m*/*z* 200, intensity threshold: 5000) of the most intense ions.

Raw data were analyzed using the MaxQuant proteomics pipeline (version 1.5.3.30) and the built-in Andromeda search engine with the human Uniprot database [22]. Carbamidomethylation was set as a fixed modification, while the oxidation of methionine as well as the acetylation of N-terminus were set as variable modifications. The search engine peptide assignments were filtered at 1% FDR. Peptides with a minimum length of seven amino acids and a maximum of two miscleavages were further processed. Peptides were matched between runs. Label-free quantification (LFQ) was performed for all samples, using razor peptides to define groups of peptides and unique proteins belonging to the razor peptides groups with the largest matches. The software Perseus (version 1.5.6.0) was used to collate LFQ data, perform imputation and log_2_ transformation of data prior to the input of data into custom R script for further analysis using version 4.0.4 [23].

### 2.4. Metabolomics

The medium was changed 24 and 4 h before harvesting cells for metabolome analysis using pulsed stable isotope-resolved metabolomics [24] in tandem with absolute quantitative gas chromatography (GC-MS). Cells were labeled prior to harvest with a medium containing either 14 mM U-^13^C-glucose with 2 mM ^12^C-glutamine for glucose tracing or 2 mM U-^13^C-glutamine with 14 mM U-^12^C-glucose for glutamine tracing. Harvested cells were washed with HEPES buffer (140 mM NaCl, 5 mM HEPES, pH 7.4) containing labeled or unlabeled glucose/glutamine and quenched by adding 50% ice-cold methanol containing 2 μg/mL cinnamic acid (Merck, Darmstadt, Germany) as an internal control. Polar metabolites were extracted as previously described [25]. Metabolite analysis was performed on a 1D gas chromatograph coupled to a time-of-flight mass spectrometer (GC-ToF-MS, Pegasus IV-ToF-MS-System, LECO, St. Joseph, MI, USA), and samples were handled with an auto-sampler (MultiPurpose Sampler 2 XL, Gerstel, Germany). The samples were injected in split mode (split 1:5, injection volume 1 μL) in a temperature-controlled injector (CAS4, Gerstel, Germany) with a baffled glass liner (Gerstel, Germany). The following temperature program was applied during sample injection: initial temperature of 80 °C for 30 s followed by a ramp with 12 °C/min to 120 °C, and a second ramp with 7 °C/min to 300 °C and final hold for 2 min. Gas chromatographic separation was performed on a Restek Rxi-5ms column (Restek, Bad Homburg, Germany). Helium was used as a carrier gas with a flow rate of 1.2 mL/min. Gas chromatography was performed with the following temperature gradient: 2 min heating at 70 °C, first temperature gradient with 5 °C/min up to 120 °C and hold for 30 s; subsequently, a second temperature increase of 7 °C/min up to 350 °C with a hold time of 2 min. The spectra were recorded in a mass range of *m*/*z* = 60 to 600 mass units with 20 spectra/s at a detector voltage of 1650 V. The GC-MS chromatograms were processed with ChromaTOF software (LECO, St. Joseph, MI, USA). Mass spectra data were extracted from the Chromatof software using proprietary methods and processed using the software tool MetMax version 1.0.1.12 (MPIMP, Golm, Germany) [25] and in-house custom R scripts.

### 2.5. Free Nucleotide Quantification

Intracellular nucleotide pools were quantified in cell lines from extracted polar metabolites using direct-infusion MS. The sample preparation was previously described [26]. The measurement of nucleotides was performed on a TSQ Quantiva triple quadrupole mass spectrometer (Thermo Fisher Scientific Waltham, MA, USA) coupled to a Triversa Nanomate nanoESI ion source (spray voltage: 1.5 kV, head gas pressure: 0.5 psi), and argon was used as a collision gas (pressure: 1.5 mTorr). The FWHM Resolution for both Q1 and Q3 was set at 0.7. Data acquisition was run for 3 min per sample, using a cycle time of 3.3 s and total acquisition of 55 SRM scans for each nucleotide in negative mode. Samples were injected in 3 technical replicates, and the median of the sum of the two transitions for each of the nucleotides were calculated for analysis. The software Xcalibur (Thermo Fisher Scientific Waltham, MA, USA) was used to manually check the quality of each measurement; scans which presented poor signal acquisition or stalling in scanning were omitted. Data were further processed with an OpenMS package, and using in-house developed R scripts.

### 2.6. Immunofluorescence Imaging

Cells were seeded on circular glass slides in 6-well plates at 1 × 10^5^ cells per well. The media were exchanged for treatment with 1× PBS or 1 mM L-GA and incubated for 24 h. After treatment, the media were removed, and the glass slides were washed with 500 μL 1× PBS. PBS was aspirated and 500 μL of 100% ice-cold acetone was added to each well containing the slides for 10 min. The slides were then washed 3 times with PBS. 50 μL Cytopainter Phalloidin-iFluor 488 (10×) (ab176753, Abcam, Cambridge, UK) and 50 μL of Deoxyribonuclease I, Alexa Fluor 594 (1 mg/mL) (D12372,Thermo Fisher Scientific Waltham, MA, USA) was added to each well and incubated for 20 min in the dark. Following the staining, glass slides were washed 3 times with 1× PBS and stored at 4  °C before scanning with a Keyence bz-x700 microscope (Keyence, Osaka, Japan) at 594 nm and 488 nm. Exposure times were maintained between treatment conditions. Images were processed with ImageJ 1.8.0 software. For unstained cells, cells were imaged via phase contrast microscopy at 10× magnification on a Nikon Eclipse TS2 microscope (Nikon, Tokyo, Japan).

### 2.7. Cell Cycle Analysis

Cells were seeded at 1.5 × 10^6^ cells per 10 cm plate and allowed to settle for 24 h. Following treatment, cells were trypsinized centrifuged for 5 min at 250× *g*, 4 °C. The supernatant was aspirated and discarded. The cell pellet was resuspended in 1ml ice-cold 1× PBS before centrifuging for 5 min at 250× *g*, 4 °C. The cell pellet was resuspended with 1 mL of ice-cold 70% ethanol in a drop-wise fashion while gently vortexing. Cells were fixed overnight at 4 °C. Following fixing, cells were resuspended in 500 μL 1× PBS and 25 μL RNase A solution (10 mg/mL) (Qiagen, Hilden, Germany). Following incubation, 15 μL of propidium iodide (PI) (1 mg/mL) (Thermo Fisher Scientific Waltham, MA, USA) and incubated in the dark for 15 min at 4 °C. PI Stained cells were transferred to fluorescence-activated cell sorting (FACS)-tubes (Sigma-Aldrich, St. Louis, MO, USA), for Fluorescence-activated cell sorting (FACS) on a BD LSR Fortessa II analyzer (Becton Dickinson, Franklin Lakes, NJ, USA) running BD FACSDIVA software (v6.0). Cell doublets were excluded by the evaluation of the propidium iodide area (PI-A) versus propidium iodide width (PI-W) density plot. Cell debris was excluded by analyzing the forward scatter area and side scatter area (FSC-Area versus SSC-Area). The distinct cell cycle phases: sub-G1, G1, S and G2/M, were determined in a propidium iodide histogram. Further data analysis was performed with the FlowJo-X software (v10.0.7r2).

### 2.8. Reactive Oxygen Species Quantification

Cells were seeded at 2.5 × 10^4^ cells per well in a black-walled transparent bottom 96-well plate (Corning, Corning, NY, USA). Cells were allowed to attach overnight then washed once with 1× buffer provided with the DCFDA/H2DCFDA—Cellular ROS Assay Kit (ab113851, ABCAM, Cambridge, UK). Cells were stained with 25 μL DCFDA in 1× Buffer for 45 min at 37 °C in an incubator. Staining media was removed and media was replaced with the drug treatment. Following 4 h incubation plates were scanned at Ex/Em: 485/535 nm on a Spectramax iD microplate reader (Molecular Devices, San Jose, CA, USA). Relative fluorescence was determined relative change to control after background subtraction.

### 2.9. iCelligence Cell Growth Analysis

Cells were seeded at 1.5 × 10^6^ cells per 10 cm plate and allowed to settle for 24 h. Cells were optimized on the iCelligence™ system (ACEA Biosciences, Inc., San Diego, CA, USA) for a seeding density which provided optimal log-phase growth after 24 h from seeding. For both IMR-32 and SH-SY5Y a cell seeding density of $1.5\times 10^{4}$ per well was found to be optimal. The iCelligence™ 16-well plate was measured for background impedance and electrical contacts were checked. Cells were resuspended to a concentration of 1.5 × 10^5^ cells/mL in fresh media, 100 μL of cell suspension was added to each of the 16-wells of the iCelligence™ plate. Each of the wells were then adjusted to 200 μL with fresh media. The plate was left for 30 min at room temperature. The plate was then inserted into the iCelligence™ device and cultivated in an incubator at 37 °C, 5% CO_2_, 21% O_2_ and 85% relative humidity. After 24 h, the media were exchanged in each well to correspond with treatments. Impedance readings were taken in triplicate from each well every 15 min. Data were analyzed with iCelligence™ RTCA DA software (version 1.0.0.1304).

### 2.10. Statistics Analysis

Statistical analyses were performed using the RStudio Desktop software (version 4.2.1). All data are presented as mean ± SEM (standard error of the mean) unless specified otherwise. Student *t*-tests were employed for analysis between groups; normality was not tested for. To test the significance between multiple groups, ANOVA and Tukey HSD post hoc tests were performed. The significance levels were set at ns p≥0.05; * p≤0.05; ** p≤0.01; *** p≤0.001; **** p≤0.0001.

## 3. Results

### 3.1. L-Glyceraldehyde Inhibits Cell Growth in Neuroblastoma Cells and Induces Apoptosis

The working concentration of glyceraldehyde was reported to be between 0.5 and 2 mM in vitro and 0.5–1 g/kg in vivo [4,5,6,9]. We aimed to confirm the working concentration of L-GA in neuroblastoma cells and characterize the rate of growth inhibition. Five neuroblastoma cell lines were treated with a range of L-GA concentrations (0–10 mM) for 24–96 h, and a WST-1 assay was performed. The IC50 of L-GA was found to be in the range of 262–1005 μM after 24 h treatment (Table 1).

Additionally, neuroblastoma cell lines IMR-32 and SH-SY5Y and the non-cancer fibroblast cell line (VH-7) were treated with a range of L-GA concentrations (0–1.5 mM) for 24 h prior to cell counting. VH-7 cells did not achieve a 50% decrease at the highest L-GA concentration used in this experiment. IMR-32 and SH-SY5Y both achieved a similar 50% decrease in cell number at L-GA concentrations: 0.76 mM and 0.55 mM, respectively (Figure 1A). These results are similar to the L-GA concentrations found to inhibit glycolysis during experiments conducted in the 1940s, whereby sub 1 mM L-GA inhibited glycolysis in muscle extracts [4].

In order to assess the effect of L-GA on cell proliferation, 1 mM L-GA was applied to VH-7, IMR-32 and SH-SY5Y for up to 48 h (Figure 1B). Viable cell counts were measured every 24 h. Following 24 h of incubation with L-GA, cell counts were reduced approximately 10-fold relative to the control in IMR-32 and SH-SY5Y. The inhibition of cell growth persisted over the 48 h incubation. The fibroblast cell line, VH-7, showed a less extreme response to L-GA treatment than the neuroblastoma cell lines; however, the cells grew slower in the presence of L-GA.

Flow cytometry was performed to examine whether apoptosis is induced by L-GA. The sub-G1 fraction of a panel of L-GA-treated cells was measured following 72 h exposure (Figure 1C, Appendix A). All neuroblastoma cell lines except CLB-GA presented a significant increase in the sub-G1 phase, alluding to an apoptotic response. VH-7 did not show an increase in the sub-G1 fraction in response to L-GA. We concluded that L-GA is active against neuroblastoma cells at concentrations ≤1 mM. Sub-G1 cell cycle fractions were indicative of apoptotic response and motivated us to perform further multi-omic experiments to identify the mode of L-GA action on cell death.

### 3.2. Proteomic Analysis of L-GA Treated Neuroblastoma Cells Reveals an Increase of Proteins Associated with Oxidoreductase Activity

At present, there are no reports in the literature of the effect of L-GA on the proteome. In order to garner a broad picture of the effect of L-GA, LC-MS shotgun proteomics was performed in five neuroblastoma cell lines (SH-SY5Y, IMR-32, BE(2)-C, GI-M-EN, SK-N-AS) and one non-cancerous fibroblast cell line (VH-7). Cells were challenged with 1 mM L-GA for 24 h. Proteins were extracted from cell lysates and label-free quantification was performed. Label free quantities (LFQ) of 3551 proteins were measured, of which 244 were annotated to be associated with 6 pathways of interest. Hierarchical clustering (Figure 2A), of the difference in log2(LFQ) (L-GA-PBS) for each cell line was performed. Specifically, we aimed to investigate the proteins associated with apoptosis, cell cycle, glycolysis, nucleotide metabolism, oxidoreductase and the TCA cycle. SH-SY5Y and IMR-32 showed the strongest increase in proteins of the oxidoreductase family following L-GA treatment. GI-M-EN and SK-N-AS also showed increased oxidoreductase activity, whereas VH-7 and BE(2)-C produced a less pronounced response in the same pathway.

Ontological enrichment was performed on all proteins in which the difference in log2(LFQ) (L-GA-PBS) was >1 or <−1 in order to define which molecular and biological processes were perturbed upon L-GA treatment in all cell lines (Figure 2B) [27]. The top five term names, defined by lower adjusted *p*-value ≤ 0.05 of the GO:Molecular function, GO:Biological process and REACTOME were selected for graphical representation. Up-regulated, defined by a difference of >1, GO subsets showed the enrichment of oxidoreductase activity, cellular organization and autophagy. The down-regulated process, defined as a difference <−1, was found to be enriched in the cell cycle and cell division processes, inter-conversion of nucleotide di- and tri-phosphates, and extracellular matrix constituents. We selected SH-SY5Y and IMR-32 from the hierarchical clustering data to compare against the fibroblast cell line, VH-7. It was found that, in general, proteins of the oxidoreductase family showed the greatest increase in IMR-32 and SH-SY5Y when exposed to L-GA. HMOX1 was found to be four times higher in IMR-32 (Figure 2C). We concluded that L-GA causes modulation to oxidative stress response, nucleotide biosynthesis and cell cycle proteins. Further experiments aimed to characterize the metabolic mode of action on these processes.

### 3.3. L-GA Depletes Nucleotides Pools, Causes Cytoskeletal Aberrations, Cell Cycle Arrest and Inhibition of Growth

Glycolytic inhibition via L-GA has been extensively reported and challenged since the 1930s [2,3,4,5,28]. It was hypothesized that L-GA could deplete ATP, redox associated nucleotides, and DNA/RNA pools. Initial experiments followed previous protocols, whereby IMR-32, SH-SY5Y and VH-7 were treated with 1 mM L-GA for 24 h. Cells were harvested, and metabolites were extracted and enriched for nucleotides. In neuroblastoma cells, DNA and RNA nucleotide pools were significantly depleted following 24 h incubation with 1 mM L-GA (p≤0.0001) (Figure 3A). VH-7 showed less reduction in DNA and RNA nucleotide pools compared to the neuroblastoma cell lines.

ATP, NAD+, NADH, NADP+ and NAPDH with associated ratios of the cell were measured in VH-7, IMR-32 and SH-SY5Y (Figure 3B). In both neuroblastoma cell lines, the ATP ratio significantly decreased (p≤0.01), this was not observed in VH-7. NAD(P)+ and NADP(H) were decreased in neuroblastoma cells with a slight decrease in the NAD+/NADH ratio. However, it is to be noted that NAD+/NADH ratios were calculated when both co-factors are strongly depleted. NAD+ was depleted more in IMR-32 and NADP+ in SH-SY5Y. L-GA caused an imbalance in the NAD+/NADH ratio at concentrations above 0.5 mM, in a similar fashion to total nucleotide pool depletion. The phosphorylation potential (ATP + 1/2 ADP/ATP + ADP + AMP) [29] of the cell is severely inhibited at 1 mM L-GA (Appendix B
Figure A1A,B). Taken together, it is apparent that the cell experiences redox stress and a reduction in the ability to perform phosphorylation reactions.

We suspected that the changes in cell morphology shown by phase-contrast microscopy (not shown) was a result of a compromised actin cytoskeleton. In order to assess the effect of L-GA on the cytoskeleton, cells were stained simultaneously for F-actin and G-actin. Briefly, cell lines IMR-32, SH-SY5Y and VH-7 were treated with 1 mM L-GA for 24 h. Slides were imaged via fluorescent excitation and emission (ex/em) 493/517 nm for phalloidin (F-actin) and 590/617 nm for DNase I (G-actin) (Figure 3C, Appendix B
Figure A1H). Image analysis showed increased G-actin fluorescence and altered F-actin structures in neuroblastoma cells. VH-7 fibroblast cell line, actin structure appeared to be less compromised upon L-GA treatment. Given that cellular division is a process in which the regulation of the cytoskeleton is essential, it was assessed whether cellular division is also compromised.

Cells were treated with 1 mM L-GA for 24 h before harvesting and stained with propidium iodide and analyzed by flow cytometry. Cells were gated into G1, S or G2/M cell cycle phases, and the distribution was calculated in percentage. Cell lines VH-7, IMR-32, and SH-SY5Y were investigated as shown in Figure 3D. IMR-32 showed a significant decrease in cells in the S phase (p≤0.01) and an increase in the G2/M phase. SH-SY5Y cells observed an increase in the S-phase and an increase in the G2/M phase (p≤0.05). In VH-7, there was a minute increase in the G2/M phase and a significant decrease in the S phase in response to L-GA (p≤0.05).

The observation of cell cycle arrest leads to the hypothesis that essential nutrients are not available to the cell. The regulation of the cell cycle is tightly controlled. Checkpoint kinases receive input from nutrient levels to initiate entry and exit from cell cycle phases [30,31,32,33]. Consequently, we aimed to assess the effect of L-GA on glycolysis and nucleotide metabolism.

### 3.4. L-GA Rapidly Inhibits Glycolysis and Nucleotide Metabolism

Previous research showed that L-GA inhibited the glycolysis of T98G and HEK293 cells [34]. Specifically, L-GA treatment resulted in the inhibition of the flux of carbon from glucose into metabolites downstream of aldolase, which utilizes glyceraldehyde-3-phosphate (GA3P) as a substrate. Furthermore, L-GA was shown to be more effective at inhibiting glycolysis than the D-GA chiral isomer. Given this, the aim was to show that neuroblastoma cells are also susceptible to glycolytic inhibition via L-GA. To measure the flux of carbon through metabolic pathways, we measured the change in metabolite pool sizes in cooperation with pulsed stable isotope resolved metabolomics (pSIRM) to derive the quantity of ^13^C-labeled metabolites [35]. We aimed to characterize the effect of L-GA in the early stages of treatment time on the central carbon metabolism and nucleotide metabolism.

Initially, we assessed how rapidly nucleotides and NAD(P)/H species deplete following L-GA treatment. IMR-32 and SH-SY5Y were treated for 1 h and 8 h, and nucleotides were extracted and measured as previously described (Figure 4A). SH-SY5Y cells appeared to increase ATP, NAD(H) and NADP(H) after one hour—with NAD+ showing a strong increase—yet early nucleotide intermediates start to decrease (R5P, PRPP). In IMR-32, nearly all nucleotides were slightly decreased after 1 hr treatment, except for AICAR and NADPH. Following 8 h treatment, almost all nucleotides were severely decreased in both cell lines. In addition, we found that D-GA was less effective at depleting nucleotides than L-GA even after 24 h in both cell lines (Appendix B
Figure A1D). Given this, we conducted pSIRM after 1 h treatment to capture early metabolic effects.

Cell lines IMR-32 and SH-SY5Y were treated for 1 h with 1 mM L-GA or 1 mM D-GA and labeled with ^13^C-glucose for 10 min within the treatment time. Extracted metabolites were analyzed by GC-MS for their quantity and amount of ^13^C-label. In both cell lines, the quantity of glyceric acid (GAc) was significantly increased upon both L-GA and D-GA treatment (Figure 4B). In IMR-32, glycerol (Glyc) did not show the same increase, suggesting that glyceraldehyde is converted mostly to GAc. However, in SH-SY5Y, more Glyc was produced. In IMR-32 and SH-SY5Y, there was no increase in glyceric acid-3-phosphate (3PGA), suggesting that GAc cannot be phosphorylated further. In SH-SY5Y, there was an increase in Glycerol-3-phosphate (Glyc3P), converse to IMR-32. In the first step in glycolysis, glucose (Glc) to glucose-6-phosphate (G6P) showed reduced G6P quantities in both cell lines only when L-GA was present. Similarly, L-GA induced greater pyruvate (Pyr) accumulation than D-GA. In IMR-32 TCA cycle intermediates were slightly decreased in both L-GA and D-GA conditions. In SH-SY5Y citrate (Cit), succinate (Suc) and malate (Mal) were slightly increased, and fumarate (Fum) was slightly decreased under L-GA treatment.

Figure 4C shows ^13^C-labeled quantities (^13^C-Peak area/$1\times 10^{6}$ cells) of glycolytic intermediates following 10 min labeling with ^13^C-glucose. In both cell lines, G6P was depleted, however, only with L-GA treatment. Labeled quantities of 3PGA were decreased in both cell lines upon L-GA treatment, although not with D-GA. Label incorporation into Glyc3P was significantly depleted in IMR-32, although only with L-GA; SH-SY5Y did not show the same degree of depletion.D-GA did not deplete Glyc3P-labeled quantities in either cell line. Labeled quantities of Lac were reduced significantly by L-GA and D-GA, although only in IMR-32. Pyr showed decreased labeled ^13^C quantities; however, this was more apparent in IMR-32.

Given that glycolysis was decreased, we examined whether the cell would fuel the TCA via glutaminolysis. Cell lines IMR-32 and SH-SY5Y were treated for 1 h with 1 mM L-GA or 1 mM D-GA and labeled with ^13^C-glutamine for 30 min within the treatment time (Appendix B
Figure A1E). In both cell lines, labeled ^13^C-quantities of TCA intermediates were increased when applying L-GA. In IMR-32, D-GA did not produce the same response as L-GA; however, in SH-SY5Y, D-GA also increased labeled TCA intermediates. We suspected that L-GA induced glutaminolysis promotes oxidative stress via increased ROS, producing mitochondrial activity.

### 3.5. L-GA Induces ROS Production, Inhibiting Nucleotide Synthesis and Arresting Cell Cycle Progression

We aimed to quantify the generation of reactive oxygen species in L-GA-treated cells. This was prompted by the observation of dysregulated NAD species, and the increase in the response of the oxidoreductase pathway in the proteomics data. We suspected that the co-application of an antioxidant, N-acetyl-cysteine (NAC) would relieve ROS-dependent cell stress.

Cell growth analysis was performed over a 72 hr time period with treatments of 1× PBS, 1 mM L-GA, 100 μM H_2_O_2_, 1 mM L-GA + 5 mM NAC, 100 μM H_2_O_2_ + 5 mM NAC and 1× PBS + 5 mM NAC. Figure 5A shows cell counts sampled every 24 h in duplicate for IMR-32 and SH-SY5Y. For both cell lines, cell growth is inhibited with 1 mM L-GA. Upon supplementation with 5 mM NAC, L-GA-treated cells showed less severe and delayed growth inhibition. The 100 μM H_2_O_2_ stopped cell growth in IMR-32 cells within 24 h to below detectable cell concentrations. The addition of NAC recovered the effect of H_2_O_2_ on cell growth. SH-SY5Y showed markedly less inhibition of cell growth in response to H_2_O_2_ compared to IMR-32. In both cell lines, NAC only partially recovered cell growth in L-GA-treated cells, whereas NAC negated the effect of H_2_O_2_, indicating that oxidative stress induced by L-GA is only one facet of its action. Furthermore, applying L-GA to cells and replacing with fresh media did not result in cellular recovery (Appendix B
Figure A1C). To clarify this, we aimed to show that the inhibitory effect of L-GA was in partial relation to the production of ROS.

ROS generation can be monitored by the fluorometric analysis of the oxidation of 2′,7′ –dichlorofluorescin diacetate (DCFDA) by ROS into 2′, 7′–dichlorofluorescein (DCF). We assessed whether L-GA induces ROS generation and if this can be rescued by NAC. Cell lines IMR-32 and SH-SY5Y were treated with 1× PBS, 1 mM L-GA, 100 μM H_2_O_2_, 1 mM L-GA + 5 mM NAC, 100 μM H_2_O_2_ + 5 mM NAC and 1× PBS + 5 mM NAC. Cells were incubated for 4 h with each treatment condition before DCFDA staining and fluorescence reading. Figure 5B shows the fluorescence intensity after treatment normalized to PBS-treated cells. In both cell lines, 100 μM H_2_O_2_ induced a significant increase in fluorescence in IMR-32 and SH-SY5Y (p≤0.001). L-GA induced an increase in fluorescence in both cell lines. Upon treatment with 5 mM NAC, the fluorescence of 100 μM H_2_O_2_ treated cells decreased significantly (p≤0.01). Similarly, there is a reduction in fluorescence of L-GA-treated cells in the presence of NAC. From these data, we concluded that 1 mM L-GA induces an increase in ROS. As we had previously found L-GA to deplete nucleotides, we questioned whether nucleotide pools are sensitive to oxidative stress and if NAC could recover depletion.

Nucleotides were measured after 24 h treatment with 1 mM L-GA, 100 μM H_2_O_2_ and 5 mM NAC, and in the combinations shown in Figure 5C. As found previously, there was a significant reduction in nucleotide pools following L-GA treatment (p≤0.0001). Upon 100 μM H_2_O_2_ treatment, nucleotides were significantly reduced in both cell lines (p≤0.0001). The co-application of L-GA and NAC resulted in a significant increase in nucleotide pools relative to L-GA alone (p≤0.05). The application of NAC to 100 μM H_2_O_2_ treated cells also significantly increased the nucleotide pools, relative to 100 μM H_2_O_2_ alone (p≤0.01).

We examined ATP, NAD+, NADH, NADP+ and NAPDH with associated ratios, following ROS treatment (Figure 5D). The ATP concentration was reduced significantly upon L-GA and H_2_O_2_ treatment (p≤0.01) in both cell lines. The addition of NAC partially remedied the L-GA and H_2_O_2_ dependent depletion of ATP. L-GA caused a decrease in NAD+ and NADH with an increase in the NAD+/NADH ratio in both cell lines. We found that in IMR-32 cells, 100 μM H_2_O_2_ caused a slight increase in the NAD+/NADH ratio. The addition of NAC to L-GA and H_2_O_2_ treated cells reduced the NAD+/NADH ratio, with the strongest increase in NADH levels. Ultimately, the availability of NAD(P)+ and NAD(P)H was compromised in both cell lines. Although nucleotide pools—and ATP in particular—were depleted in both cell lines by H_2_O_2_, the growth of SH-SY5Y cells persisted. This suggests that the induction of ROS production and nucleotide depletion are not fully responsible for cell growth inhibition by L-GA. Nucleotide depletion is likely to be a concerted phenotype in the multi-modal action of L-GA.

Following the findings of the partial recovery of nucleotide pools via NAC, we hypothesized that the cell cycle block induced by L-GA could be recovered by NAC. IMR-32 cells and SH-SY5Y cells were challenged with 1 mM L-GA or 1 mM L-GA + 5 mM NAC for 24 h (Figure 5E). Cells were harvested and stained with propidium iodide and analyzed by flow cytometry. As found previously, L-GA caused a significant increase in the G2/M phase (p≤0.01) in IMR-32 cells. The co-application of NAC significantly relieved the G2/M block relative to L-GA alone (p≤0.05). In SY-SY5Y cells, L-GA caused a significant decrease in the G1 phase and a significant increase in the S phase (p≤0.01). Upon co-application with NAC, we found a significant increase in the G1 phase (p≤0.001) and a significant decrease in the S phase (p≤0.05) relative to L-GA alone. In summary, L-GA induces ROS generation, dysregulates NAD(P)+ and NAD(P)H availability, depletes nucleotides, and causes cell cycle arrest. The co-application of NAC partially relieves these cell states induced by L-GA. NAD+ and NADH generation is tightly linked to central carbon metabolic processes.

### 3.6. N-Acetyl-Cysteine Partially Restores Glycolytic Function of L-GA-Treated Cells

Cell lines IMR-32 and SH-SY5Y were treated for 1 h or 16 h with 1× PBS, 1 mM L-GA, or 1 mM L-GA + 5 mM NAC. Cells were incubated with U-^13^C-glucose for 10 min, and metabolites were analyzed (Figure 6).

For SH-SY5Y cells, we found a decrease in ^13^C-labeled quantities of Gly3P, Lac, and Pyr following 1 h treatment with L-GA. The co-application of L-GA+NAC significantly increased ^13^C-labeled quantities in Glyc3P; however, no effect was observed with Lac or Pyr. After 16 h treatment, Lac showed a stronger response to L-GA treatment, which was mitigated with NAC co-application. Labeled quantities of Glyc3P and Pyr were slightly reduced upon L-GA application relative to PBS, with L-GA+NAC showing a marginal increased labeled quantity relative to L-GA alone.

For L-GA-treated IMR-32, ^13^C-labeled quantities for Glyc3P, Pyr and Lac were decreased after 1 hr treatment. The addition of NAC at this time point partially mitigated L-GA-dependent inhibition at Pyr and Lac. After 16 h treatment, Pyr- and Lac-labeled quantities remained lower than the control for L-GA cells. The addition of NAC partially restored labeled quantities in a similar fashion to 1 h with Pyr and Lac.

In summary, we find that NAC partially recovers the L-GA-mediated inhibition of glycolysis, particularly at Lactate. We suspect this to be a result of the NAD+/NADH cycling governed by glyceraldehyde-3-phosphate dehydrogenase (GAPDH), Lactate dehydrogenase (LDH) and Glycerol-3-phosphate dehydrogenase (GPD). The sensitivity of GAPDH to oxidative stress is well documented, although LDH and GPDH are not known to be ROS sensitive; however, both enzymes require NADH. We observed a heterogeneous response to L-GA between cell lines, particularly at Glyc3P. IMR-32 appeared to respond to the NAC-mediated recovery of glycolytic much earlier than SH-SY5Y; this may be due to the difference in rate of the L-GA metabolism in each cell line. It is likely that L-GA acts as a direct inhibitor of glycolysis and secondly as a ROS generator with latent indirect effects which are partially recovered by NAC. To further address this bi-modal effect, we examined post-translational modification to the proteome.

### 3.7. Phosphoproteomic Analysis Reveals Cell Cycle-Dependent Kinases Are Perturbed by L-GA

A phosphoproteomic approach was performed in order to derive information about which signaling pathways are modulated by L-GA. In addition, we aimed to show that affected pathways are relieved via NAC application. IMR-32 cells were treated with 1× PBS, 1 mM L-GA or 1 mM L-GA + 5 mM NAC for 4 h before harvesting. Enriched phosphoproteomic samples were measured via LC-MS. Intensities of peptide sequences with phosphorylation sites were annotated with associated gene names and kinases using the PhosphoSitePlus database [36]. The fold change in phosphopeptide intensities (L-GA/PBS) was calculated, mapped to the parent protein, and plotted against the −log10 (*p*-value) (Figure 7A). Phosphopetides were also annotated with predicted kinases that phosphorylate the peptide. Phosphopetides phosphorylated by kinases ERK2, AuraA;CDK1, CK2A1 GSK3A, MAPKAPK2 were found to be the top five regulated following L-GA treatment. When assessing the proteins mapped to the phosphopetide, we found the strongest response from PARVA, TP53, DTD1, MAPT, and CDC25B. Although not annotated with an associated kinase, we found strong up-regulation of ATN1 and down-regulation of CRAMP1L.

In order to discern which phosphopeptides were most significantly affected by the addition of NAC, we examined the relative difference on phosphopeptide intensities between L-GA and L-GA+NAC treatment (Figure 7B). We found the largest difference in kinases GSK3A, PKACA, ERK1, CK1D, and CDK2. Therefore, phosphopeptides regulated by these kinases, modulated by L-GA, were relieved by NAC. Although not annotated with a kinase, we found NCOA3, RANGAP1, and GTPBP1 to be the strongest responders to NAC following L-GA treatment.

We found that L-GA exhibits a post-translational effect on cells at the phosphoproteomic level. The addition of NAC appears to relieve pathways modulated by L-GA. Interestingly, the top modulated pathway, ERK2, has a broad effect on cellular metabolism, cell cycle regulation, and cell adhesion. This modulation correlates with the the cell cycle analysis, actin staining, and metabolomics experiments performed previously.

## 4. Discussion

Homeostasis is essential for cancer cell survival. We have shown that by challenging neuroblastoma cells with—a long-time presumed—glycolytic inhibitor, homeostasis is severely affected. Subsequently, a novel mechanism is presented for L-GA action, in which multiple facets of the metabolism are dysregulated and not limited to glycolytic inhibition (Figure 8). Analysis of nucleotide metabolism, cell cycle arrest, and (phospho)proteomics presented L-GA-induced oxidative stress. This mode of action of L-GA was previously unreported in neuroblastoma cells, and is of particular significance in understanding its therapeutic potential. The function of L-GA as a glycolytic inhibitor has dominated the focus of its previous research; however, the mechanism has not been characterized. Employing the pSIRM strategy showed that L-GA decreases the carbon flow into metabolites that require NAD(H) as a co-factor. Through the integration of GC and DI mass spectrometry data sets, it was indicated that L-GA breaks the cycling of cytosolic NAD+/NADH. This breakage may impact the redox state of the cell, thereby inhibiting metabolism. Glyceraldehyde dehydrogenase (GAPDH) produces cytosolic NADH via the metabolism of glyceraldehyde-3-phosphate to 1,3-glyceric acid biphosphate. GAPDH possess a cysteine-rich active site, thereby rendering it sensitive to ROS [37]. We suspect that GAPDH activity may be impacted by L-GA directly by delivering a smaller amount of the substrate glyceraldehyde-3-phosphate, and indirectly as elevated ROS levels via L-GA inhibiting GAPDH. D,L-glyceraldehyde has been shown to raise intracellular peroxide levels, dysregulate NAD(P)/NAD(P)H ratios, acidify the cell and inhibit glycolysis—at more than double the concentration used in this paper—in islet cells via non-mitochondrial pathways. This phenomenon causes oxidative stress which is remedied with NAC application [20,38]. Hydrogen peroxide has been shown to inhibit GAPDH activity. This results in reducing carbon flow into lower glycolysis intermediates, and upper glycolysis intermediates accumulate [39]. The reduction in carbon flow complements our data with respect to L-GA, although, Van Der Reest and colleagues reported the depletion of pyruvate levels, where we observed pyruvate accumulation in the time frames employed. Furthermore, we found that D-GA has a lesser effect on label incorporation into Lactate and Glycerol-3-phosphate than L-GA and does not reduce label incorporation into glyceric acid-3-phosphate. D-GA may be metabolized by ALDO(A/C) to produce fructose-1-phosphate, which can be used to re-fuel glycolysis, and L-GA may produce sorbose-1-phosphate, which inhibits hexokinase [40]. L-GA did not cause a notable accumulation of upper glycolysis intermediates as would be expected with GAPDH inhibition, which may be a result of hexokinase inhibition. We found ALDOC protein levels in SH-SY5Y to be higher than IMR-32, which may give reasoning to the lower response to D-GA (Appendix B
Figure A1I). This was complemented by nucleotide analysis, in which D-GA did not show the same level of nucleotide depletion as caused by L-GA in IMR-32.

We found that IMR-32 and SH-SY5Y respond to L-GA differently, particularly in the early stages of L-GA treatment; it appears that SH-SY5Y is more sensitive to NADPH depletion, as well as nucleotide deletion. This may be due to higher levels of ARKB1, which consumes NAPDH in the production of glycerol from L-GA (Appendix B
Figure A1I). IMR-32 tended to be more sensitive to glycolytic inhibition and NAD(H) depletion. This may be due to the metabolic rates of the cells and their relative dependency on subsequent metabolic pathways. Specifically, we found varying levels of aldehyde dehydrogenase and aldo-keto reductase between cell lines (Appendix B
Figure A1I). This is certainly an issue of consideration when discussing the L-GA efficacy and mechanism between cell types.

The application of L-GA and H_2_O_2_ to neuroblastoma cells induced a depletion of nucleotide pools. Literature on the effect of ROS on the nucleotide intermediate metabolism is sparse. Although much is known about the effect of ROS on the depletion of ATP production and DNA damage, the work here provides novel insights into how ROS affects nucleotide intermediates. The application of NAC in the presence of H_2_O_2_ and L-GA indicates that nucleotide synthesis is indeed ROS sensitive, although it may not be necessary for cell survival, as in the case of sustained SH-SY5Y cell growth following H_2_O_2_ application. We suspect that L-GA-derived ROS and nucleotide depletion alone does not induce apoptosis, but rather the sum of all L-GA modes of action result in cell death. Despite this, further work is required to identify the specific enzymes which are most susceptible to ROS. It is unclear whether it is a direct action of ROS on nucleotide metabolizing enzymes or whether it is a result of signaling pathway perturbation. Enzymes associated with nucleotide biosynthesis RNR and TYMS are known to be ROS sensitive [41,42]. However, our results suggest that additional nucleotide biosynthesis enzymes are subject to modulation via ROS, directly or indirectly. Through the measurement of the product/substrate ratios of nucleotide intermediates, it was evident that SH-SY5Y and IMR-32 respond differently to oxidative stress. Curiously, with early intermediates such as SAICAR, they displayed accumulation upon ROS exposure (Appendix B
Figure A1F). The characterization of all nucleotide intermediates via DI-MS/MS is expected to precisely reveal additional L-GA targets within the nucleotide biosynthesis pathway.

Albonzinia et al. (2022) showed that *MYCN* amplified neuroblastoma cells exhibit a high demand for cysteine [43]. *MYCN* amplification results in the over-expression of anti-oxidant genes to maintain the redox balance, which is stressed due to depleted cysteine from high *MYCN* levels. *MYCN* levels in IMR-32 are amplified (100 *MYCN* copies) compared to a gain (3 *MYCN* copies) in SH-SY5Y. Given that our observations showed IMR-32 to be more sensitive than SH-SY5Y to H_2_O_2_, it is possible that the basal state of IMR-32 is under more oxidative stress due to amplified *MYCN*.

Phosophoproteomic data highlighted kinase pathways, which were modulated via L-GA treatment. Specifically, GSK3β and MAPKAPK2 axes were identified by MAPT and PARVA. MAPKAP2 is known to regulate the G2/M transition in response to DNA damage via Cdc25B/C phosphorylation [44]. Given that L-GA depletes nucleotides, it is likely that DNA damage ensues, and appropriate responses are initiated. Indeed, we found Cdc25B/C phosphorylation to be up-regulated upon L-GA treatment, giving credence to MAPKAPK2-14-3-3 mediated cell cycle arrest and apoptosis (Appendix B
Figure A1G). Furthermore, we found p53 associated with AurA and CDK1 kinases to be up-regulated in response to L-GA, which is indicative of DNA damage response and G2/M arrest [45,46]. AurA has been implicated in high-risk neuroblastoma and there are current efforts to validate AurA inhibitors [47]. It would be of clinical interest to examine the efficacy of AurA inhibitors and L-GA co-application in neuroblastoma models.

Many chemotherapeutics induce ROS, pushing the cancer cell to lethal limits [48]. Doxorubicin, Daunorubicin and cisplatin induce high levels of ROS [49,50]. Nucleotide analogs such as 5-fluoropyrimidine produce lower levels of ROS [51]. L-GA acts in a similar manner, however, possessing significantly lower toxicity. In a complementary context, our data are supported by the findings of Uetaki et al. (2015). The authors showed that high levels of ascorbic acid induce strikingly similar effects in MCF7 human breast adenocarcinoma and HT29 human colon cancer cells as L-GA does in neuroblastoma cells. Ascorbic acid induces HMOX1 overexpression, depletes dNTPs, and inhibits glycolysis by NAD+/NADH dysregulation. However, the global depletion of nucleotides and their intermediates via ascorbic acid was not shown [52].

The targeting of cancer cell metabolism by the glycolytic inhibitors 2-DG and BrPyr have shown to be ineffective or toxic due to narrow therapeutic windows [53,54,55,56]. Yet in light of their molecular mechanism, BrPyr has been shown to induce ROS increase and the activation of GSK3β as found with L-GA treatment [57,58,59]. In combination with MAPKAPK2 and TP53 up-regulation, glycolytic inhibitors are shown to induce DNA damage responses and cell cycle arrest.

L-GA is of particular clinical interest due to its low toxicity, being relatively inexpensive and easy to synthesize. When bracketed in a manner similar to chemotherapeutics, the therapeutic concentration of L-GA required is relatively high [53,60,61]. Our data suggest that maintenance of a working concentration may not be necessary, as we found that L-GA-treated IMR-32 cells could not recover after L-GA was removed (Appendix B
Figure A1C).

For conventional regimens such as vincristine [O], cisplatin [P], etoposide [E], and cyclophosphamide [C], OPEC, dosages are in the micromolar range. Yet, patients experience severe side effects ranging from hair loss, appetite suppression, nausea, and increased chance of resistance [62]. Due to the high molarity of L-GA required for potential therapy, it is unlikely to translate into a mono-therapy. However, L-GA has scope to be used as an adjunct to provide more efficacy to chemotherapeutic regimens.

## 5. Conclusions

This study aims to characterize the mechanism by which L-GA inhibits neuroblastoma cell growth. It was found that L-GA targets multiple facets of metabolism in neuroblastoma cells. We believe that L-GA inhibits glycolysis, resulting in reactive oxygen species production and increased oxidative stress on the cell. This stress directly or indirectly—through signaling—decreases nucleotide biosynthesis and initiates cell cycle arrest and apoptosis. Through the co-application of an antioxidant, L-GA activity was decreased, highlighting oxidative stress as a key component of the L-GA action. This finding was supported by flow cytometry and proteomic analysis.

## Figures and Tables

**Figure 1 cancers-16-01664-f001:**
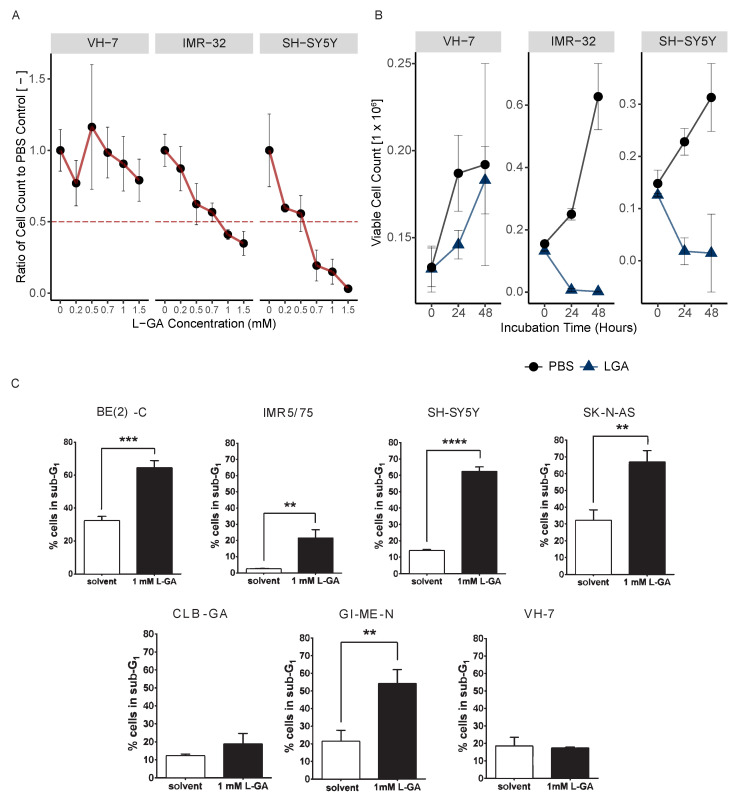
L-Glyceraldehyde inhibits cell growth in neuroblastoma cells. (**A**) Cell lines VH-7, IMR-32 and SH-SY5Y cells were incubated with varying concentrations of L-GA for 24 h after seeding. Viable cell counts were taken for each L-GA concentration via the trypan exclusion method and normalized to 0.0 mM control. The 50% reduction in cell count is denoted by a dotted red line with errors bar representing the ± SEM (*n* = 3). (**B**) Cell lines were incubated with 1 mM L-GA (blue) or PBS (black) for 48 hrs after seeding and settling for 24 h. Viable cell counts were taken every 24 h via the trypan exclusion method. Error bar representing the ± SE of the mean (*n* = 3). (**C**) Cell lines BE(2)-C, IMR5/75, SH-SY5Y, SK-N-AS, CLB-GA, GI-M-EN and VH-7 were treated with L-GA for 72 h before flow cytometry analysis. Sub-G1 cells were gated, and a two-sided T-test was performed (*n* = 3). ** p≤0.01; *** p≤0.001; **** p≤0.0001.

**Figure 2 cancers-16-01664-f002:**
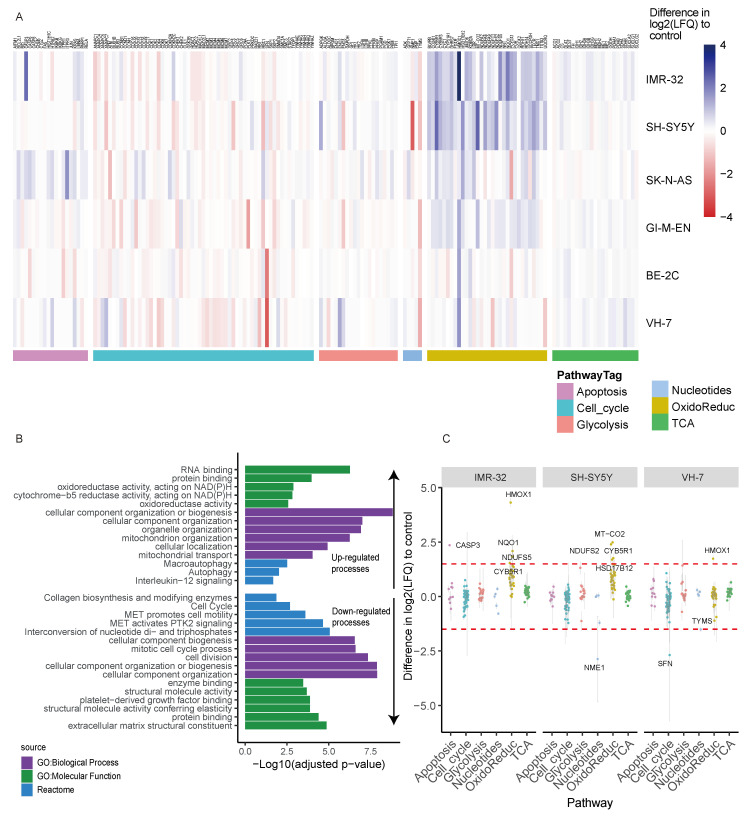
Proteomic analysis of L-GA treated neuroblastoma cells. (**A**) Proteomic analysis of cell lines: BE-2C, GI-M-EN, VH-7, SH-SY5Y, IMR-32 and SK-N-AS treated for 24 h with 1 mM L-GA or PBS and subjected to LC-MS proteomic analysis. Hierarchical clustering of cell lines show the clustering of pathways between cell types. Data represent the difference in Log_2_ label-free quantity Log_2_(LFQ) of L-GA-treated cells relative to PBS-treated cells, *n* = 3 for each cell line. (**B**) GO enrichment analysis of data filtered by >1 and <−1 difference Log_2_(LFQ) of L-GA-treated cells relative to PBS control in all cell lines. The top five term names, defined by a lower adjusted *p*-value < 0.05, of the GO:Molecular function, GO:Biological process and REACTOME are presented. (**C**) Proteins from (**A**) were filtered by >1.5 and <−1.5 difference Log_2_(LFQ) of L-GA-treated cells relative to PBS in IMR-32, SH-SY5Y and VH-7. Filtered proteins are highlighted by a red dotted line. Proteins that show differences outside of this line are labeled by text.

**Figure 3 cancers-16-01664-f003:**
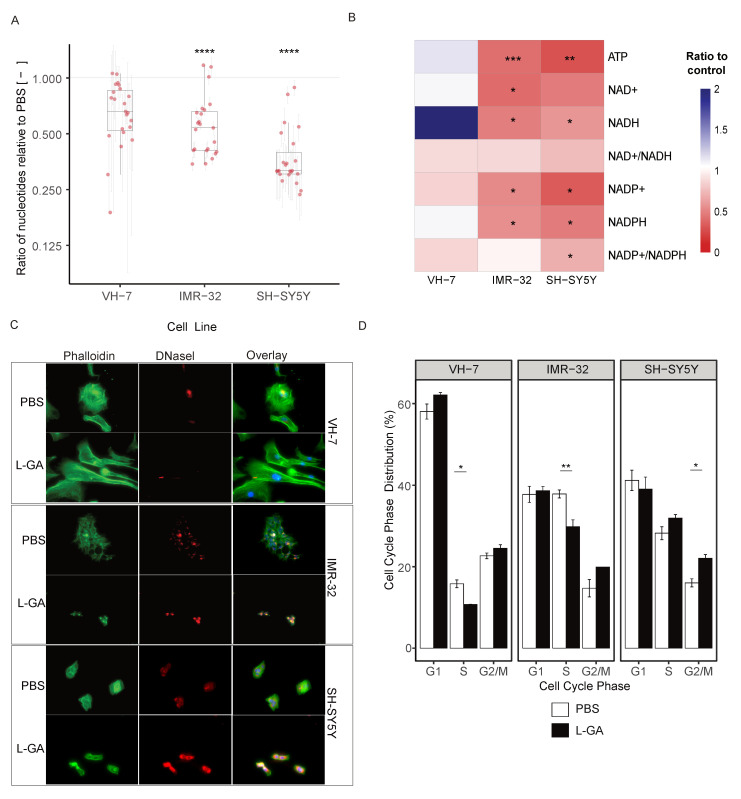
L-GA causes cytoskeletal aberrations and cell cycle arrest within 24 h. (**A**) Cell lines VH-7, IMR-32 and SH-SY5Y were treated with PBS or 1.0 mM LGA 24 h. Nucleotides were extracted and measured by DI-MS/MS. Boxplots represent the mean ratio in each DNA and RNA nucleotide intensity normalized to PBS (*n* ≥ 3). Error bars represent ±SEM of each nucleotide measured. *p*-values were calculated using a two-tailed *t*-test. (**B**) Cells were treated as in (**A**); intensities of ATP, NAD+, NADH, NADP+ and NADH with associated ratios were calculated and normalized to PBS (*n*≥ 3). *p*-values were calculated using a two-tailed *t*-test. (**C**) Cell lines VH-7, IMR-32 and SH-SY5Y were fixed to glass slides and stained with DAPI and Phallodin-iFluor(F-) or Deoxyribonuclease I (G-) actin probes then imaged at 40× magnification using Keyence Bz-x700 microscope. Images are exemplary of 3 independent slides. (**D**) SH-SY5Y, IMR-32 and VH-7 were treated with 1 mM L-GA or PBS for 24 h and subjected to flow cytometry analysis. Cells were gated for cell cycle phases G_1_, S, and G_2_/M. The mean of the percentage of cells in each phase was calculated (*n* = 3). Error bars represent ± SEM. Two-tailed *t*-tests were conducted in each phase between treatments. * p≤0.05; ** p≤0.01; *** p≤0.001; **** p≤0.0001.

**Figure 4 cancers-16-01664-f004:**
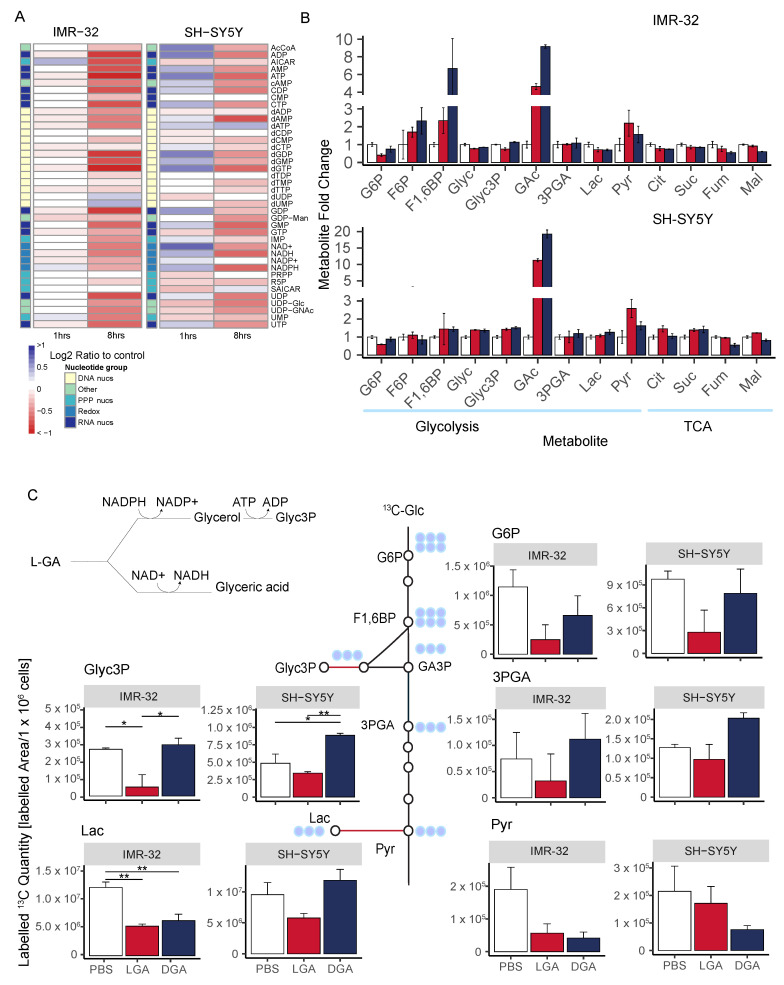
L-GA selectively and rapidly inhibits glycolysis. (**A**) Cell lines IMR-32 and SH-SY5Y were treated with 1 mM L-GA for 1 or 8 h. Nucleotides were extracted and measured by DI-MS/MS. The fold change in nucleotide intensity relative to PBS was calculated for representation by heatmap (*n* = 4). (**B**) Polar metabolites were extracted from IMR-32 and SH-SY5Y cell lines following the treatment of 1 mM L-GA or 1 mM D-GA for 1 h. Samples were processed via GC-MS and analyzed to attain the abundance of metabolites relative to PBS. Bar charts represent the metabolite mean fold change, error bars represent ± SEM (*n* = 3). (**C**) Cell lines IMR-32 and SH-SY5Y were treated with 1 mM L-GA or 1 mM D-GA for 1 h and labeled for 10 min with U-^13^C-glucose within the treatment time. Bar charts depict the mean labeled ^13^C-quantity (labeled peak area/$1\times 10^{6}$ cells) of depicted metabolites. Error bars represent ± SEM (*n* = 3). *p*-values were calculated using ANOVA and Tukey post hoc tests. * p≤0.05; ** p≤0.01.

**Figure 5 cancers-16-01664-f005:**
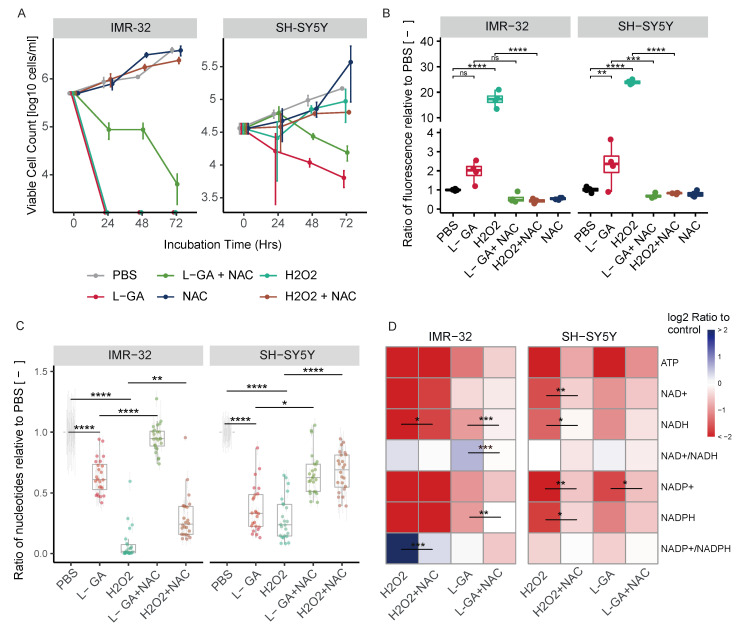
L-GA induces oxidative stress which is linked to cell growth, nucleotide synthesis and the cell cycle. (**A**) IMR-32 and SH-SY5Y were treated for 72 h with 1 mM L-GA, 100 μM H_2_O_2_ and 5 mM NAC, in the combinations shown. Viable cell counts were taken every 24 h. Error bars represent the ± SEM (*n* = 3). (**B**) IMR-32 and SH-SY5Y were treated for 4 h in combination as in (**A**). ROS species were quantified by measuring the fluorescence from DCFDA. Box plots show the ratio of fluorescence relative to the PBS control (*n* = 4). *p*-values were calculated using ANOVA and Tukey post hoc analyses. (**D**) Cells were treated as in (**C**); the intensities of ATP, NAD+, NADH, NADP+ and NADH with associated ratios were calculated and normalized to PBS (*n* = 4). *p*-values were calculated using ANOVA and Tukey post hoc analyses. (**E**) Cell cycle analysis was performed in IMR-32 and SH-SY5Y cells following treatment for 24 h with PBS, 1 mM L-GA or 1 mM L-GA + 5 mM NAC. Cells were gated into phases for ANOVA and Tukey post hoc analyses on the cell cycle distributions (*n* = 3). ns p≥0.05; * p≤0.05; ** p≤0.01; *** p≤0.001; **** p≤0.0001.

**Figure 6 cancers-16-01664-f006:**
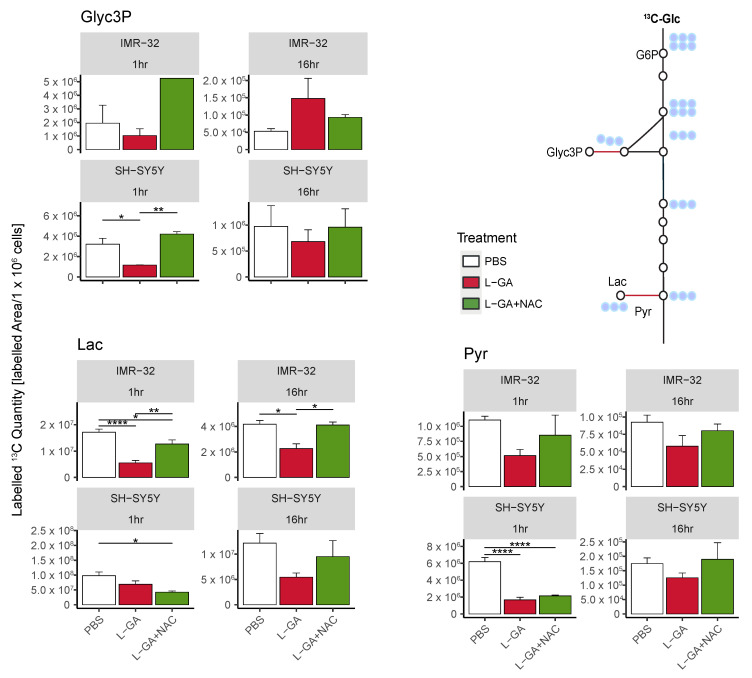
The antioxidant N-acetyl-cysteine partially restores the metabolic function of L-GA-treated neuroblastoma cells. Polar metabolites were extracted from IMR-32 and SH-SY5Y cell lines following treatment of 1 mM L-GA or 1 mM L-GA + 5 mM NAC for 1 and 16 h and 10 min labeling with U-^13^C-glucose. Samples were processed via GC-MS and analyzed to attain the peak area and label incorporation to calculate the labeled area for each metabolite. Bar charts depict the mean labeled ^13^C quantity (labeledarea/$1\times 10^{6}$ cells) (*m*/*z* +3). Error bars represent ±SEM, (*n* = 3–4, except IMR-32: Glyc3P, 1hr L-GA+NAC (*n* = 1)). *p*-values were calculated using ANOVA and Tukey post hoc tests. * p≤0.05; ** p≤0.01; **** p≤0.0001.

**Figure 7 cancers-16-01664-f007:**
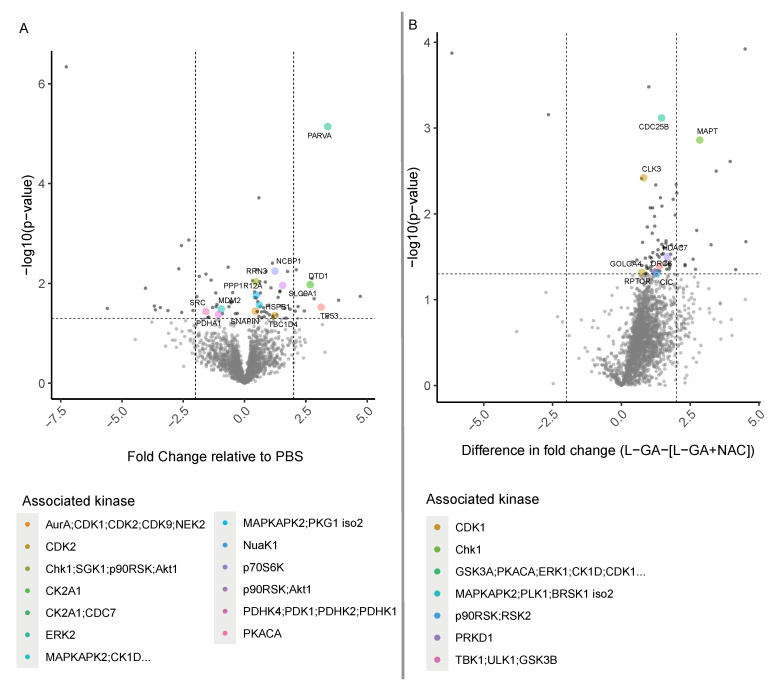
Phosphoproteomic analysis of L-GA-treated cells. (**A**) IMR-32 cells were treated for 4 h with 1 mM L-GA, 1 mM L-GA + 5 mM NAC and PBS and harvested for phosphoproteomic analysis (*n* = 4). Phosphopeptides were measured by LC-MS and fold changes were calculated for L-GA/PBS and L-GA+NAC/PBS. Significant phosphopeptides were found by ANOVA followed by Tukey post hoc tests, colored by the associated kinase. Phosphopeptide sequences were annotated with a protein name and associated kinase. (**B**) Fold change data between L-GA and L-GA+NAC treatment were passed to a two-tailed *t*-test. Significant peptides were annotated with protein names and associated kinases and highlighted in red (*n* = 4).

**Figure 8 cancers-16-01664-f008:**
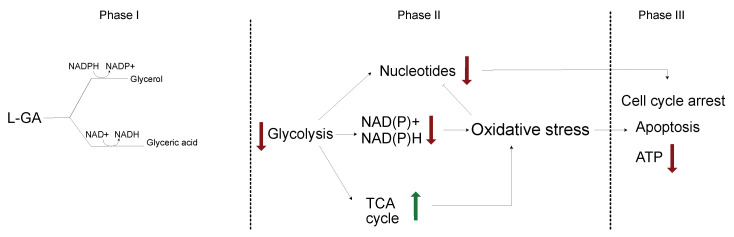
Proposed mechanism of L-GA on cellular metabolism. L-GA is rapidly converted to glyceric acid, favoured over glycerol metabolism, consuming NAD+. At adequate concentrations of L-GA, glycolysis is inhibited resulting in further decrease of ATP and NAD derived co-factors. This results in an increase of oxidative stress which co-currently up-regulates the TCA cycle, leading to ROS production and further oxidative stress. Nucleotides are depleted as a facet of cell stress and glycolysis is further inhibited. As a sum of parts, the cell enters cell cycle arrest and initiates apoptosis. Red arrows indicate a decrease in pathway/metabolite and green arrows indicate an increase in the pathway/metabolite.

**Table 1 cancers-16-01664-t001:** IC50 values of a panel of neuroblastoma cell lines and a fibroblast cell line treated with L-GA via WST assay over 96 h treatment time.

Cell Line	*MYCN*-Status	IC50 24 h	IC50 48 h	IC50 72 h	IC50 96 h	Delta IC50
BE(2)-C	amplified	1005	285	137	447	469
SH-SY5Y	gain	660	783	94	300	460
SK-N-AS	single copy	1001	622	228	1069	730
CLB-GA	single copy	995	98	110	103	104
GI-M-EN	single copy	262	196	110	214	196
VH-7	-	1042	963	301	244	638

## Data Availability

The mass spectrometry proteomics data have been deposited to the ProteomeXchange Consortium via the PRIDE [63] partner repository with the data set identifier PXD048695. Other data will be made available upon request.

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
