# Peer review of "L-Glyceraldehyde Inhibits Neuroblastoma Cell Growth via a Multi-Modal Mechanism on Metabolism and Signaling"

_cancers, 2024, doi:10.3390/cancers16091664_

Round 1
Reviewer 1 Report
Comments and Suggestions for Authors
1. why authors see difference (variation) in cell ration in VH-7 cells with different concentration of the L-GA? please explain
2. Why there are less viable cells in VH-7 at 24 hours compared to PBS, and almost similar at 48hours (fig 1B), please explain
3. please provide flow data instead of graphical representation (Fig 1C)
4. Fig 3C, is L-GA effecting actin polymerization?
5. please correct Fig 3C and 3D legend
Author Response
- why authors see difference (variation) in cell ration in VH-7 cells with different concentration of the L-GA? please explain
L-GA did affect VH-7 growth to a non-negligible degree. Admittedly, the variation of the cell counts for VH-7 was rather large, at all concentration points. We have increased the contrast on the error bars and given the discrete x-axis values on Fig1A for clarity of the variation. Erroneously, the manuscript contained figure1A showing standard deviations rather than SEM as described in the caption. This has been updated appropriately.
- Why there are less viable cells in VH-7 at 24 hours compared to PBS, and almost similar at 48hours (fig 1B), please explain
As with the previous question, L-GA did affect VH-7 growth, the cells grew slower in the presence of L-GA. At the 48hr time point the variation in the cell count has been made clearer by increasing error bar contrast in Fig1B. VH-7 grew a lot slower than the neuroblastoma cell lines, also in PBS conditions, as in apparent by comparison of the y-axis between all cell lines.
- please provide flow data instead of graphical representation (Fig 1C)
Flow cytometry data has been provided as supplementary material in file supplementary_material_01.zip.
- Fig 3C, is L-GA effecting actin polymerization?
From the F-actin and G-actin staining, we saw a trend towards more G-actin signal in the presence of L-GA. Furthermore, we saw from the F-actin stain, that the actin cytoskeleton is structurally different in L-GA treated neuroblastoma cells. However, this experiment was done in a steady state condition and dynamic polymerization cannot be resolved. In order to assess the dynamic nature of actin polymerization, a time resolved imaging strategy would be required, therefore from our data we do not conclude that actin polymerization is affected by L-GA, however the cytoskeletal structure is. We have revised the text at lines 337-346 to make this clearer.
- please correct Fig 3C and 3D legend
The correction has been applied in the caption of figure 3.
Reviewer 2 Report
Comments and Suggestions for Authors
"L-Glyceraldehyde inhibits neuroblastoma cell growth via a multi-modal mechanism on metabolism and signaling" by Forbes et. al., studies the mechanisms by which L-Glyceraldehyde (L-GA) can inhibit cell cycle progression in different neuroblastoma cell lines. The authors mostly present proteomics data and gene ontology analysis besides flow cytometry to conclude that L-GA causes cell cycle arrest to inhibit neuroblastoma cell growth. My concerns are below:
1. The authors must show in L-GA treated neuroblastoma cells, the expression of phosphorylated Histone-3 at Serine-28, these cells are in the M phase and thus this protein can be used as a marker to screen M-phase cells. So immuno- fluorescence with phospho Histone-3 antibody can help determining cells in G2 or M phase along with western blots.
2. Include more microscopy data from cells to show L-GA treated cells have increased production of ROS.
Author Response
"Please see the attachment."

Round 2
Reviewer 2 Report
Comments and Suggestions for Authors
The authors have reasonably addressed my concerns. I would recommend accepting the revised manuscript for publication.